# How do German General Practitioners Manage Long-/Post-COVID? A Qualitative Study in Primary Care

**DOI:** 10.3390/v15041016

**Published:** 2023-04-20

**Authors:** Beatrice E. Bachmeier, Salome Hölzle, Mohamed Gasser, Marjan van den Akker

**Affiliations:** 1Institute of Pharmaceutical Biology, Goethe-University, 60438 Frankfurt am Main, Germany; 2Institute of General Practice, Goethe University, 60590 Frankfurt am Main, Germany; 3Department of Family Medicine, Care and Public Health Research Institute, Maastricht University, 6229 ER Maastricht, The Netherlands; 4Department of Public Health and Primary Care, Academic Centre of General Practice, KU Leuven, 3000 Leuven, Belgium

**Keywords:** Long-/Post-COVID, qualitative study, primary care

## Abstract

Background: Many patients with ongoing complaints after a SARS-CoV-2 infection are treated in primary care. Existing medical guidelines on how to diagnose and treat Long-/Post-COVID are far from being comprehensive. This study aims to describe how German general practitioners (GPs) deal with this situation, what problems they experience when managing such patients, and how they solve problems associated with the diagnosis and treatment of Long-/Post-COVID. Methods and Findings: We conducted a qualitative study and interviewed 11 GPs. The most commonly described symptoms were ongoing fatigue, dyspnea, chest tightness and a decrease in physical capacity. The most common way to identify Long-/Post-COVID was by exclusion. Patients suffering from Long-/Post-COVID were generally treated by their GPs and rarely referred. A very common non-pharmacological intervention was to take a wait-and-see approach and grant sick leave. Other non-pharmacological interventions included lifestyle advices, physical exercise, acupuncture and exercises with intense aromas. Pharmacological treatments focused on symptoms, like respiratory symptoms or headaches. Our study’s main limitations are the small sample size and therefore limited generalizability of results. Conclusions: Further research is required to develop and test pharmaceutical and non-pharmaceutical interventions for patients with Long-/Post-COVID. In addition, strategies to prevent the occurrence of Long-/Post-COVID after an acute infection with SARS-CoV-2 have to be developed. The routine collection of data on the diagnosis and management of Long-/Post-COVID may help in the formulation of best practices. It is up to policymakers to facilitate the necessary implementation of effective interventions in order to limit the huge societal consequences of large groups of patients suffering from Long-/Post-COVID.

## 1. Introduction

In 2020, SARS-Coronavirus Type 2 (severe acute respiratory syndrome coronavirus type 2, SARS-CoV-2) spread rapidly throughout the world, causing a pandemic and affecting all aspects of life, including health, the economy, and community life [1].

Similar to other infectious diseases, some patients have continuing complaints following an acute infection with SARS-CoV-2. If the complaints continue for four weeks or more, the condition is referred to as Long-COVID syndrome, or post-acute sequelae of COVID-19. If the symptoms persist for longer than 12 weeks, the condition is referred to as Post-COVID syndrome [2]. In this paper, we will not differentiate between Long- and Post-COVID and persisting symptoms following an acute infection with SARS-CoV-2 will be referred to as Long-/Post-COVID syndrome [3].

Long-/Post-COVID syndrome is a condition that follows COVID-19, is classified as an illness-associated health problem in the ICD-10 (international statistical classification of diseases and related health problems 10th revision), and is becoming increasingly common worldwide. At present, its prevalence can only be estimated. An umbrella review comprising 23 reviews and 102 original papers reported prevalence estimates in these manuscripts ranged from 7.5% to 41% in non-hospitalized adults, 2.3–53% in mixed-adult samples, 37.6% in hospitalized adults, and 2–3.5% in predominantly non-hospitalized children [4].

Risk factors for the development of Long-/Post-COVID are, among others, advanced age, female sex, hospitalization and deteriorating symptoms during the course of the COVID-19 disease [5].

Studies on the persistence of post-acute symptoms with a follow-up of less than 12 weeks revealed that fatigue, dyspnea, arthralgia and a dry or productive cough occurred most frequently. Persisting gastrointestinal symptoms such as diarrhea, anorexia, nausea, stomach pain, and dysphagia were less frequently observed. The symptoms that most frequently persisted between 12 weeks and 6 months were again fatigue, dyspnea and a dry or productive cough. Many patients also suffered from thoracic pain (prevalence of all symptoms are reviewed in [6]). The persistence of related neurological symptoms was also relatively common, notably including smell disorders, taste disorders, and rhinitis [7]. It was also found that females were at higher risk of symptom persistence [6,8].

At present, the diagnosis of Long-/Post-COVID is not reflected in any specific laboratory parameters [9].

Although guidelines are gradually being developed for the diagnosis and treatment of Long-/Post-COVID, much remains unclear, and existing guidelines for general practitioners and medical specialists on how to diagnose and treat Long-/Post-COVID are far from comprehensive. We conducted a qualitative study to find out how German general practitioners (GPs) deal with this situation, what problems they experience when managing such patients, and how they solve problems associated with the diagnosis and treatment of the condition.

## 2. Methods

### 2.1. Study Design and Participants

This paper presents the results of a qualitative study of how German GPs diagnosed and treated Long-/Post-COVID patients in their practices, that was carried out at Goethe University, Frankfurt am Main, Germany. We performed semi-structured individual interviews with maximum variation sampling to capture GPs from both sexes, from urban and rural areas, and with differing amounts of work experience. GPs were either recruited through the research networks of a number of Institutes of General Practice in Germany, or via direct contact to GPs.

### 2.2. Informed Consent

Before any study-specific activities were undertaken, potential participants were sent participant information and an informed consent form via email. Written informed consent was obtained by means of a dated signature by the participant and was repeated orally at the beginning of the interview.

### 2.3. Interview Guide

BB and MvdA developed a semi-structured interview guide so that the researcher asked specific predetermined questions, while still ensuring participants had the chance to discuss issues they felt were important. The interview guide was discussed with an interprofessional group of qualitative working researchers (including colleagues of different ages and different sexes). The adapted guide was tested in a pilot interview and proved feasible and comprehensible.

### 2.4. Data Collection

All interviews were performed by the same researcher (BB). We originally planned to conduct 10 to 15 in-depth telephone interviews with GPs. The interviews were performed between 11 March and 11 May 2022, digitally recorded, and afterwards transcribed verbatim. In order to protect the identity of study participants, names and other identifying details were changed during transcription.

### 2.5. Data Analysis

Coders (MG and MvdA) read all transcripts to familiarize themselves with the data. In our codebook, we started by classifying the topics referred to in our interview guide, thus facilitating the initiation of data analysis. During the coding process, codes were altered and added when appropriate [10]. MG and MvdA coded the data independently. Coding was continuously compared and discussed after each 2–3 interviews. Discrepancies in findings were resolved by discussion, which involved a third researcher (BB) in case of disagreement. To ensure intersubjective reproducibility and comprehensibility, the results were presented to and discussed with the co-authors SH and BB. Subsequently, citations were clustered into subtopics.

### 2.6. Data Protection

All confidential data will be kept on password-protected computers for ten years. The data can only be accessed by members of the research team, and all personal data were pseudonymized. Personal data were replaced by a letter code. During transcription, audio-recordings were anonymized, with all identifiable information removed prior to the coding procedure. The audio-recordings were deleted immediately after transcription.

## 3. Results

A total of 11 GPs were interviewed (Table 1), about half of whom were female. On average, they were 47.5 years old and had 20.3 years of professional experience. Most had specialized in general practice, while 2 had specialized in internal medicine. About one third worked in a single-handed practice, and about two thirds in a group practice.

### Definition and Epidemiology of Long-/Post-COVID Syndrome

According to GPs, a diagnosis of Long-/Post-COVID was always preceded by a SARS-CoV-2 infection. Patients that showed no signs of improvement during the six to eight weeks following the infection were suspected of having Long-/Post-COVID. If patients did not get well after 3 to 6 months, GPs considered the state of the disease to be chronic, whereby it should be noted that the definition of chronic Long-/Post-COVID disease varied from GP to GP.


*No, more like eight weeks. I think you really have to wait eight weeks.*
(Ms. E)


*Well, when they come for the first time, three or four weeks after the infection, then I say everything’s still ok, but when it goes on for a month or two, then I do say that it’s probably Long-COVID.*
(Ms. I)


*And in four or five female patients—after nine months, they still showed nothing: that is to say after nine months there was still no improvement. I would then refer to them as having Long-COVID. And in one case, it was recognized as being an occupational disease.*
(Mr. K)

The question how many Long-/Post-COVID cases GPs were currently treating was difficult to answer because they did not have an exact overview of the extent to which their patients had recovered, particularly in the case of those that hadn’t consulted their GP for a long time. Nonetheless, estimates of how many Long-/Post-COVID patients were receiving treatment in particular practices varied between 10 and 50. According to the GPs, the percentage of patients developing Long-/Post-COVID syndrome ranged from 1% to 30%.


*Well, of those, I calculated that overall this year I had 190 cases of Corona of which I coded 40 as U099, meaning Long-COVID or Post-COVID. And of those, there were a handful, that is to say about five maybe, right?*
(Mr. H)


*So 10, 20 Long-COVID cases in the practice that I guess I know about. And it’s probably about two of those, meaning about 10 percent.*
(Ms. I)

## 4. Diagnosis

### 4.1. Symptoms and Complaints

The reported spectrum of Long-/Post-COVID symptoms differed in number and quality. The majority of GPs described symptoms such as ongoing fatigue, dyspnea, chest tightness and a decrease in physical capacity, while some also reported rarely occurring exceptional symptoms. Exertional dyspnea and performance reduction were particularly detrimental to the patient’s day-to-day productivity and overall health.


*And they have dyspnea a lot, when they exert themselves, tightness in the thorax, a cough—those are the complaints that patients have at the moment.*
(Ms. G)


*He’s exhausted and has exertional dyspnea, so when he exerts himself physically, his physical activity is limited and he increasingly experiences shortness of breath—and then often excess mucus and pressure on his chest, so a bit like pulmonary symptoms. That’s on the one hand, and in addition to all the symptoms of exhaustion.*
(Ms. F)

Furthermore, patients struggled with sleep apnea, reduced concentration and cognitive difficulties. Migraines and headaches, gastritis, a persistent cough, increased sensitivity to pain, non-specific joint and muscle pain, general irritability, dizziness, and weight loss were also mentioned. Interestingly, the symptoms of olfactory dysfunction that were frequently reported during the first wave of SARS-CoV-2 were rather rare from the second wave onwards.


*Well, I’ve experienced patients whose symptoms are mostly respiratory, as well as those that primarily have gastrointestinal issues, but in the end they just feel exhausted all the time, with difficulties in being productive and in concentrating—kind of like being a completely different person that they don’t recognize at all and with illnesses they’ve never experienced before.*
(Mr. K)

### 4.2. Diagnostic Procedures

Long-/Post-COVID was characterized by a certain pattern of symptoms, hence the most common way to identify it was by exclusion. The only precondition was a previous SARS-CoV-2 infection. Moreover, no specific lab parameters gave an indication of it and no particular cluster of symptoms corresponded to the syndrome.


*We often carry out general health check-ups, as one does, by measuring blood pressure, listening to sounds in the body, (oxygen) saturation, whereby I normally measure saturation for the patient’s sake, so they can see it’s not too low.*
(Ms. J)

The GP’s typical examination generally involved taking a medical history. Additionally, a small blood count was measured to rule out thyroid disease or a vitamin deficit and to check important inflammatory parameters such as d-dimer and troponin. They also frequently performed pulmonary function tests to exclude diseases with similar symptoms, such as asthma or COPD, even though in general the test results were normal.


*And we do pulmonary function tests in the practice, the results of which are generally perfectly normal. At the beginning of the Corona pandemic, we used to regularly send patients for an X-ray but stopped because the findings we got back were never remarkable in any way.*
(Ms. J)

The GPs also performed heart tests such as ECGs, or checked for rheumatic diseases. For further medical examinations, GPs also referred their patients to cardiologists and pulmonologists.


*And then, when everything’s normal there as well, if the complaints continue, we send the patient to a cardiologist for a further examination.*
(Ms. G)

### 4.3. Problems GPs Experience in Diagnosing Long-/Post-COVID

While many parameters had to be examined to rule out other diseases, GPs often complained that diagnosing Long-/Post-COVID in their practices was time-consuming, especially as the illness cannot be identified on the basis of any distinct lab parameters.


*Well, yes, there is a guideline that’s a bit of a decision aid. But it seems to me, and this is unfortunately in the literature, that there is no specific figure at the moment that would enable you to say for sure—well you know the patients have symptoms, but all the values that we measure are normal. There are none that clearly reflect these Long-COVID symptoms. I think it’s more of a clinical decision.*
(Ms. G)

In addition, waiting periods for an appointment with a cardiologist or pulmonologist tend to be long. The tight budget that German health insurance companies make available for diagnostic tests was also a problem and limited the number of tests that could be conducted. Furthermore, doctors had to pay attention to psychosomatic issues or a possible depressive component, as these could be either an indication or the result of Long-/Post-COVID. The latter was especially true of patients that were either used to performing at a high level, or already known to have mental issues, such as depressive complaints or anxiety disorder before they were infected with SARS-CoV-2.


*You have the impression that many of the patients with Long-COVID, that they were very—how would you say it? Under pressure before their COVID-illness, and very busy, you see? Very performance-oriented. And that makes it a bit difficult, of course, because you think of other things and sometimes wonder if there isn’t a depressive component behind it all. Because I mean all the symptoms clearly can make you depressed when you’re used to doing things but can’t any more.*
(Ms. G)

Communication between GPs and patients suffering from cognitive impairment or with a low level of education was also challenging at times. The circumstance that some patients appeared to be unaware of their bodily functions and did not seem to notice a change in the state of their health also complicated the diagnosis. Patients that denied the existence of Corona were also a problem because they did not want to test for COVID-19 or simply denied having Long-/Post-COVID. Sometimes, GPs had problems distinguishing between the side effects of the COVID vaccination and the course of the disease following a previous infection.


*The Corona deniers—they always play it down. They come far too late, you see? And they don’t get tested. And then I have a problem proving it and the whole legal side of things. And then of course I have the general problem in family medicine when I have people that have never learned to pay attention to what’s going on in their bodies.*
(Mr. B)


*She has a fever again and again. Now, of course, we don’t know whether the fever comes from the illness or the vaccination—the third she has had—or has she had COVID in the meantime, you see?*
(Ms. E)

### 4.4. Methods to Improve Diagnostic Investigations

An explicit Long-/Post-COVID guideline, screening tools, biological markers and multidisciplinary outpatient centers were seen as essential if diagnostic procedures were to be improved. Furthermore, a screening questionnaire with a validated scoring system would help GPs to rule out other diseases and identify Long-/Post-COVID correctly. This should be supported by biological markers to specifically indicate a previous infection.


*First of all, it would be good to have an unvarying guideline recommendation that you could rely on. I think what’s available at the moment is a bit weak. It wouldn’t be bad to have a standardized questionnaire either. And then, well the development of a guideline on it would facilitate the whole procedure of making a diagnosis a bit.*
(Ms. L)


*As I said, there’s still no kind of marker that one could rely on because everyone has normal measurements, including those from cardiac tests. And that’s what’s missing a bit.*
(Ms. G)

GPs would also welcome support from a larger number of multidisciplinary outpatient clinics, as they felt their data could be used in medical research. In general, GPs also wished for a uniform database, where data on Long-/Post-COVID could be collected and used to generate knowledge on both diagnosis and treatment. With respect to budgetary constraints, German GPs are only permitted to issue a certain number of prescriptions per patient and trimester. This restriction forced them to refer their patients to other medical specialists. GPs were therefore in favor of raising the trimestral maximum, and for multidisciplinary outpatient clinics to be established to which they could refer their patients.


*Coordination with a kind of COVID outpatient clinic, like it is with the university hospital, but just a better coordinated version, so that they, for example, have the possibility to bill for lab tests, so they don’t have to come under my budget. And, yes, that the whole thing is better coordinated and all the data are immediately used in research.*
(Mr. K)

### 4.5. Patient Characteristics Related to Long-/Post-COVID Syndrome

GPs reported gender, age and pre-existing conditions as patient characteristics related to the risk of Long-/Post-COVID. While some GPs clearly said the majority of patients with Long-/Post-COVID were female, others reported an equal number of males and females. All GPs agreed that patients with Long-/Post-COVID were generally younger, with most being 30–50 years old, but never above 70. The youngest case was a 12-year-old girl.


*That varies. I’d say that most are between 25 and 55, or 58. I’m not sure how old the female patient was, but about that. We don’t really see older patients, 60, 70.*
(Ms. G)

Remarkable differences in the symptoms in women and men have been observed. GPs described female patients as being exhausted, having headaches and an impaired sense of smell. Male patients, on the other hand, were more likely to have dyspnea and a cough. Most GPs thought the occurrence of Long-/Post-COVID was influenced by pre-existing conditions such as low resistance to stress and alcoholism. Men seemed to take longer to recover when they suffered from depression, or when they had a sensitive nature.


*No, men that took longer, they all had some kind of a mental disorder such as depression or a kind of … they are what I would call light men, men that are rather sensitive, very sensitive even. Well, kind of stressed out in advance or alcoholics.*
(Mr. B)

Furthermore, individuals whose jobs were stressful and/or required dealing with large numbers of people were more often affected by Long-/Post-COVID. Generally, both vaccinated and non-vaccinated persons were affected by Long-/Post-COVID to a similar extent. Interestingly, GPs did not see many Long-/Post-COVID cases among patients with significantly impaired oxygen saturation, even when the course of their SARS-CoV-2 infection had been severe.


*…we sent those whose illness was really severe and whose saturation was really bad for a CT scan directly and then we really saw that severe changes had taken place in their lungs. The interesting thing is that they are not the patients that sit in the consultation room every week because they‘re feeling poorly on account of Long-COVID.*
(Ms. J)

The longer patients struggled with Long-/Post-COVID, the more GPs assumed the disease was chronic. This was often the case when patients already had an underlying disease such as COPD, or peripheral arterial disease (PAD), before they were infected with SARS-CoV-2. Patients whose work was stressful were particularly likely to be unable to return to work later on.


*We have cases that become chronic. That happens. But well, they are mostly those that already have severe underlying diseases, severe COPD where the lung tissue has been significantly damaged, or PAOD, you see? Then you do see that they have problems.*
(Mr. D)


*Those that are in their early fifties, that are in the situation because of their age: Is that really incapacity for work? As I said, they are preschool teachers, most of them are preschool teachers that were probably already saying to themselves, ‘I can’t manage this any longer, with loud children for 40 h, it’s just not going to getting any better’.*
(Ms. J)

## 5. Treatment

### 5.1. Non-Pharmacological Treatment

A very common non-pharmacological intervention was to take a wait-and-see approach and grant sick leave. GPs recommended that patients give themselves time to recover. When patients were concerned about their symptoms, GPs explained to them that their situation was probably only temporary. In addition, it was helpful if key laboratory tests had been undertaken and the results had been normal.

GPs also advised their patients to modify their lifestyles by managing their diets and eating more fruits and salad, preparing more freshly cooked meals, eating less meat, and by exercising more. GPs further explained the importance of taking enough vitamin C and D through food. Smoking cessation was highly recommended, although many patients struggling with Long-/Post-COVID were non-smokers anyway. In addition, GPs advised their patients to be physically active two to three times a week and to continuously increase their level of activity in order to revascularize their small blood vessels.


*When the disease is advanced, I try to encourage patients to move so that they increasingly strain themselves, at least twice a day, in order to trigger the revascularization of small blood vessels.*
(Mr. D)

One GP suggested to patients that they should take time off to go for a relaxed walk in the forest. The idea behind such ‘bathing in the forest’ was that by connecting with their surroundings and breathing fresh air, they would reduce stress levels and raise their endurance.


*Forest-bathing is good (…) In Japan, there are even studies on it—if you do it for two hours a week (…) Well it’s like hiking, but you do it without any objective, you take your time and it makes no difference where you end up, otherwise you end up under time pressure. So you shouldn’t take the dog. You just take water with you and when you go in—you’ll need about half an hour to really be properly in there—then you’ll start sensing the forest and concentrating on listening to the birds singing etc.*
(Mr. B)

Some GPs also recommend acupuncture and exercises with intense aromas in order to help recover from any loss of smell. Other non-pharmacological interventions included rehab, breathing gymnastics and normobaric oxygen therapy.

### 5.2. Pharmacological Treatment

For Long-/Post-COVID symptoms affecting the lungs or airways, GPs prescribed drugs that are routinely used to treat such lung diseases as COPD or asthma. These were budesonide for a persistent cough and salbutamol for the treatment of airway complaints. Glucocorticoids were generally prescribed in case of fibrosis. GPs treated obstructions of the lung with a combination of Beta 2-sympathomimetics and cortisone, and recommended N-acetylcysteine, or colchicine combined with etoricoxib and prednisolone to patients suffering from pulmonary symptoms. In case of suspected viral-induced pneumonia, one GP administered 8 mg of dexamethasone.

In order to follow the course of the disease and evaluate treatment success, patients’ blood count was frequently monitored.


*Exactly, we often see an obstruction to the function of the lungs, which we then treat. We do that with a combination drug containing Beta 2-sympathomimetics and cortisone.*
(Ms. G)

For the treatment of a lost sense of smell, GPs prescribed prednisolone or glucocorticoid inhalation sprays such as Foster 200 and Trimbow.

In case of migraine, GPs gave a high dose of analgesics such as ibuprofen. For sleep disturbance, herbal drugs, vitamins and supplements were recommended.


*They take an ibuprofen, but in case of migraine I have to use high doses, and then I normally manage to stop the attack.*
(Mr. B)

### 5.3. Referrals

Patients suffering from Long-/Post-COVID were generally treated by their GPs and rarely referred to specialists or rehabilitation centers.

However, when organ or system damage was suspected, patients were referred to a specialist, or sometimes to a rehabilitation center, psychotherapist or Post-COVID clinic.

When patients were demanding, GPs sometimes involved them in their diagnostic investigations by, for example, letting them organize their own appointments with specialists or at rehabilitation centers, well aware that it might take months until they are given an appointment.


*To a certain extent you can—this sounds a bit silly—occupy patients by saying to those that are in a great hurry, ‘OK, if you want a further diagnosis’ and then giving them a referral to a neurologist or the Post-COVID outpatient clinic, knowing as I do that they will have to wait six months for an appointment.*
(Ms. J)

### 5.4. Problems in the Treatment of Long-/Post-COVID

GPs found the treatment of symptoms that are difficult to measure or to quantify such as fatigue, dizziness, loss of concentration, muscular and joint pain and unspecific lethargy to be difficult, especially in the absence of a specific guideline or therapy plan. Additional factors that further complicated the treatment and healing process were psychosomatic problems and sleeping disorders, regardless of age.

It was difficult for patients to accept the lack of well-established treatment options for Long-/Post-COVID complaints, as it made them worry that their symptoms would not improve in the near future. Younger patients, particularly young women, had a particularly hard time taking a break from work and slowing down for several months or a year to recover. Middle-aged patients started thinking about early retirement.


*Some patients—the ones that keep coming back—have difficulties accepting that there are no established therapy options.*
(Mr. C)


*…that it’s often women that can’t take time off, for whom it is out of the question when someone says to them, ‘slow down a bit for six months’, you see? They’re just people that can’t do that because they’re so bound up in their professional lives and have families to look after, where it’s just not possible not to give 100%. They also get in a kind of panic when you tell them it may take six months of a whole year until they‘re fully back on their feet again.*
(Ms. J)

Patients with high anxiety levels and those that are skeptical about a therapy’s efficacy were the most difficult to treat. This is true for ‘anti-vaxxers’, as well as vaccine supporters. Persons that work in the health system and are really exhausted, such as nursing staff working in hospitals, and in particular in COVID units, were also difficult to treat.


*Let’s just say that the illness tends to last longer in people with fairly serious underlying illnesses, or who worry a lot. It would certainly be desirable to provide them with psychotherapy, right? But I can’t afford to do that.*
(Mr. H)


*The problem of psychosomatics. And so I have to look: ‘What is the problem?’ We have problems at work at the moment, financial problems, family problems—see if it’s that. And if it’s not then see if medication works.*
(Mr. B)

GPs consider questionnaires to be a good instrument for monitoring Long-/Post-COVID complaints. However, consultations are not long enough for patients to fill in questionnaires during them.


*…no time for questionnaires. I’m not prepared to do that for patients with statutory health insurance—you see? We can’t manage that.*
(Mr. B)

The limitations to the number of prescriptions German health insurers permit GPs to write, prevented some patients from getting the required therapy. For this reason, rehab, physiotherapy and occupational therapy could not be prescribed more often.


*We recently sent a patient to rehab, and it was very good for her. But the health insurance funds strictly limit that, our ability to send people for physiotherapy or ergotherapy or whatever, you see?*
(Ms. G)

### 5.5. Factors Positively Affecting the Course of the Disease

Treatments are more likely to be successful if patients can adapt to the new situation. Persons that accept their illness and have a positive attitude, don’t appear to have Long-/Post-COVID for as long as those struggling with a negative attitude. This was also true for physically active persons that had no problems testing their limits and exercising daily, as they tended to recover from Long-/Post-COVID faster than people that required motivation.


*When the person has a positive attitude, they’re back on their feet a lot quicker.*
(Mr. H)

Furthermore, some patients appeared to accept the situation more easily if GPs confirmed that their internal organs had not been damaged and no other problems had resulted from Long-/Post-COVID.

GPs noticed that when patients received a COVID-19 vaccine while suffering from Long-/Post-COVID, their condition improved.


*And I also noticed that people with Long-COVID that were still weak after three months but were prepared to be vaccinated. When they were vaccinated, it alleviated the symptoms. I don’t know why. It has been described in the literature, so I do it, and it works somehow.*
(Mr. D)

### 5.6. How Do GPs Define Treatment Success?

The most common definition of treatment success was an improvement in physical resilience and the ability to return to work.


*That they’re healthy again, that they can work again, that they feel well again, that they can tolerate stress again.*
(Ms. F)

The enhancement of their mental and physical wellbeing, as well as their ability to concentrate, was equally important to patients. GPs considered it to be a sign of treatment success if patients did not return to them with persistent complaints.


*Let’s put it like this. Most people only come back if they still have complaints. It’s rare that anyone says, ‘I feel brilliant now’, right?*
(Ms. I)

### 5.7. Necessary Improvements in Treatment

The most common wish was that more Long-/Post-COVID centers and practices, as well as short-term rehabs (3 to 14 days) would be set up. At these institutions, psychologists, psychotherapists, physiotherapists and occupational therapists could work together to improve the physical and psychological well-being of the patient. Moreover, the possibility to refer to such specialists as neurologists, physiotherapists and psychotherapists within two weeks would relieve GPs and patients.


*Yes, I think we should collaborate with some kind of center that offers that kind of thing. I think it would make sense if some kind of institute existed where a psychologist or a psychotherapist was available, as well as physiotherapists and ergotherapists, so that they’re all in one place.*
(Ms. G)

Standardized self-assessment questionnaires for patients to fill out at home could also help monitor complaints. GPs often have no time to help patients carry out elaborate psycho-pathological tests.


*Self-assessment forms, standardized ones that you could give patients to take home with them. They could be useful in monitoring developments—that would certainly be helpful. Family practices often just don’t have the time to run extensive psycho-pathological tests.*
(Mr. C)

GPs also wanted patient information on Long-COVID to be readily available. Patients should also be informed that a healthy lifestyle can have a positive impact on the course of Long-/Post-COVID complaints and an acute Corona infectious disease, and not only on the progression of other diseases.


*Yes, in my opinion it might be good it patients were to receive more information, perhaps from various media channels, that lifestyle changes are not just good for other illnesses, but that they have a particularly positive influence on the course of Corona and Post-COVID.*
(Mr. H)

## 6. Discussion

### 6.1. Main Results and Comparison to Literature

Our study shows that the way German GPs diagnose Long-/Post-COVID syndromes is very similar, with their diagnostic strategies focusing on exclusion. The symptoms they mentioned most frequently were ongoing fatigue, dyspnea, chest tightness, a decrease in physical capacity, and to a lesser extent sleep apnea, impaired concentration and cognitive difficulties. However, we observed considerable variation in pharmacological and non-pharmacological treatments despite the publication of the German Guideline for Long-/Post-COVID syndrome [9] about a year before our interviews took place. GPs reported substantial uncertainty in both diagnosing and treating Long-/Post-COVID syndrome and wished for greater support.

Schrimpf et al. describes symptoms similar to ours, and, in agreement with our results, rarely mentioned depressive symptoms and anxiety [11].

Long-/Post-COVID is perhaps the first illness to be identified as a result of patients finding one another on Twitter, which occurred in 2020 [12]. Now, two years after the first reports of Long-/Post-COVID, the prevalence of this syndrome is still difficult to gauge. Estimates range from 2.3–53% in mixed adult samples [4]. German GPs reported that an average of 11.9 patients per practice had symptoms that persisted for 4–12 weeks and about 5.9 patients had complaints that persisted for over 12 weeks [11]. These estimates agree with the range of 10 to 50 patients in the GP practices in our study.

A scoping review on the management of Long-/Post-COVID in general practice describes that clinical uncertainty forming a diagnosis results from the multiplicity of symptoms, many of which overlap with other conditions [13]. Uncertainty regarding clinical diagnosis, etiology and the expected course of the disease was also described by O’Hare et al. [14] Furthermore, it is necessary to routinely conduct additional tests and to refer patients to specialists more often to support or refute a diagnosis of Long-COVID [14]. GPs in our sample diagnose Long-/Post-COVID by ruling out other illnesses. It would be helpful if more specific procedures were available to help differentiate Long-/Post-COVID from other diseases with similar symptoms.

Fatigue is the most frequent symptom of Long-/Post-COVID syndrome. The German Guideline on Long-/Post-COVID therefore recommends use of the Fatigue Scale (FS), the Fatigue Severity Scale (FSS) or the Fatigue Assessment Scale (FAS). If fatigue is accompanied by an intolerance of physical exertion, cognitive impairment and pain, all lasting for more than 6 months, GPs should use international diagnostic methods to check whether Chronic Fatigue Syndrome (ME/CFS) is the reason [15]. The presence of post exertional malaise (PEM) should also be ruled out [16]. Additionally, a differential diagnosis should be used to exclude the existence of other physical and mental diseases that might lead to symptoms of fatigue, as well as orthostatic intolerance and muscular fatigue.

Symptom-specific pharmaceutical treatments described in previous studies [11] were similar to ours. Non-pharmaceutical treatments generally lacked a comprehensive and holistic strategy and focused on exercise and physical rehabilitation. We would recommend a multidisciplinary approach that involves specialists from various fields.

Dyspnea and unspecified thoracic discomfort about 3–6 months after an infection with SARS-CoV-2 belong to the most common symptoms of Long-/Post-COVID. The German Guideline on Long-/Post-COVID recommends clarification through functional testing (diffusion capacity, blood gas analysis, 6-min walking tests, ergospirometry) and further cardiac diagnostics. As the pandemic is ongoing, a variety of experts and professional associations continue to vary the diagnostic tests they recommend, but lung function tests, imaging methods and an (exercise) ECG remain the most common examinations. The factors that determine the selection of diagnostic measures are the course of the COVID-19 disease, pre-existing lung conditions, a high risk of interstitial lung disease, and signs of restrictive ventilation [17]. Most of the GPs in our sample took these into consideration.

No clear pharmacological treatment yet exists for Long-/Post-COVID. Following a diagnosis in primary care, a wait-and-see policy with GP follow-up is advised in case of stable clinical symptoms [9].

At present, the main goals of Long-/Post-COVID therapies are the management of symptoms and the avoidance of disease chronification. These require good quality sleep, pain management, stress reduction, relaxation, the stimulation of personal resources and adequate coping styles (e.g., neither overdoing things, nor unnecessarily avoiding activity), as well as support through appropriate aids and the patient’s social environment.

National Institute of Health Care Excellence (NICE) guidelines for managing the long-term effects of COVID-19 published in 2020 suggest treating Fatigue Syndrome through patient self-management und controlled physical activity [18]. As the symptoms of long-/Post-COVID fatigue overlap with those of ME/CFS, similar therapy strategies are recommended, which consist of a multimodal treatment that combines cognitive behavioral therapy with moderate physical activity. In addition, specifically trained health service staff and therapists should help patients learn self-management of their symptoms [19]. On the other hand, evidence also exists that cognitive behavioral therapy should not be used to treat fatigue because of possible undesired adverse events [20], and that the use of physical activity as a therapeutic measure should be carried out in a controlled fashion to prevent post-exertional malaise (PEM) [21].

According to the German guideline, therapeutic approaches for the treatment of dyspnea should be used to relieve symptoms [9]. Patients with pulmonary hyperresponsiveness should be treated using beclometasone/formoterol (Timbow) for 6–8 weeks [22]. If interstitial or pulmonary vascular disease is suspected, patients should be referred to a specialist or to hospital.

Both the German S1 Guideline [9] and the NICE guideline [23] emphasize the importance of taking an interdisciplinary and holistic approach and ensuring continuity of care by integrating and coordinating primary care, and using rehabilitation and mental health services.

In our study we focused on the views and experiences of GPs rather than those of patients. However, some reports describe that uncertainty [24] and anxiety in patients results from doubt about the course of the disease and the fear of chronification. Such patients may need to be examined by a multidisciplinary team [24]. Some patients also describe how important it is to be listened to and treated with empathy, and to receive psychosocial support where necessary [13]. Apparently, many patients suffering from Long-/Post-COVID also experience great difficulty in receiving appropriate medical support because they (i) are unable to access healthcare, (ii) are able to access healthcare but do not receive adequate support, (iii) have extremely persistent symptoms, and (iv) feel the need for alternatives to mainstream health care [25]. Some of our GPs sympathized with these concerns and reported that all these points should be taken into consideration and addressed in order to develop efficient therapy strategies for Long-/Post-COVID.

### 6.2. Strengths and Limitations

Our study’s few limitations are the small sample size and the limited generalizability resulting from the small number of interviewed GPs. Its strengths are the in-depth information, which reflects the way in which GPs actually care for their patients with Long-/Post-COVID syndrome.

### 6.3. Recommendations

The few existing studies focusing on patient experiences unanimously show that they often experience anxiety and uncertainty. Future studies should focus on patient needs and the accessibility of care. Further research is also required to develop and test pharmaceuticals, but also to investigate non-pharmaceutical interventions to support patients with Long-/Post-COVID and to reflect upon strategies to prevent the occurrence of Long-/Post-COVID after an acute infection with SARS-CoV-2. As suggested by one of the GPs, the routine collection of data on the diagnosis and management of Long-/Post-COVID may help in the formulation of best practices and contribute to improving the treatment and support of patients. It is up to policymakers to facilitate the necessary implementation of effective interventions in order to limit the huge societal consequences of large groups of patients suffering from Long-/Post-COVID.

## Figures and Tables

**Table 1 viruses-15-01016-t001:** Characteristics of participating GPs and practices.

GP and Practice Characteristics	
Sex—N (%)	
Female	6 (54.5)
Male	5 (45.4)
Specialization—N (%)	
GP	9 (81.8)
Internal medicine	2 (18.2)
Further specialization—N (%)	
Thorax surgery	3 (27.3)
Cardiology & emergency	1 (9.1)
Psychotherapy	1 (9.1)
Practice type—N (%)	
Single, without personnel	4 (36.4)
Single, with personnel	2 (18.2)
Professional association	3 (27.3)
Group practice	2 (18.2)
Location of practice	
Large town (>100,000)	4 (36.4)
Medium-large town (20,000–100,000)	3 (27.3)
Small town (5000–20,000)	3 (27.3)
Village (<5000)	1 (9.1)
Size of practice—mean/median (range)	2482/1750 (900–5500)
<1000 patients	1 (9.1)
1000–2000 patients	5 (45.4)
2000–4000 patients	1 (9.1)
>4000 patients	2 (18.2)
Age—mean/median (range)	47.5/52 (34–61)
Years of work experience—mean/median (range)	20.3/19 (8–36)

The interviews lasted 13–37 min (mean 22.4 min).

## Data Availability

Data is unavailable due to privacy or ethical restrictions.

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
