# Peer review of "How do German General Practitioners Manage Long-/Post-COVID? A Qualitative Study in Primary Care"

_viruses, 2023, doi:10.3390/v15041016_

Round 1

Reviewer 1 Report

This study addresses a topic that is still much debated and concerns how general practitioners manage the problem of Long-/Post-COVID syndrome in the area. Semi-structured one-on-one interviews were conducted through an assessment questionnaire with maximum sampling of variation to capture general practitioners of both sexes, from urban and rural areas, and with varying amounts of work experience.

The questionnaire results would indicate that the most commonly described symptoms were continued fatigue, dyspnea, chest tightness and a decrease in physical capacity. Although Long-/Post-COVID was characterized by a certain pattern of symptoms, the most common way to identify it was the exclusion of these. Long-23/Post-COVID patients were generally treated by their primary care physicians and rarely referred to specialists or rehabilitation centers. A very common non-pharmacological intervention has been to take a wait-and-see approach and take sick leave. Other non-pharmacological interventions included lifestyle advice, exercise, acupuncture, and intense aroma exercises. Drug treatments have focused on symptoms, such as respiratory symptoms or headaches.

The conclusions underline the extensive need for further research to develop and test pharmacological and non-pharmacological interventions, and strategies to prevent the occurrence of Long-/Post-COVID after an acute SARS-CoV-2 infection. Finally, the significant weakness of the study methodology and data collection was underlined.

The manuscript is readable with great difficulty also due to the complexity of the methodological details. Given the weakness of the study method, I suggest the authors significantly reduce these parts to make it more readable to the reader

Author Response

Dear Editors,

we have carefully revised the manuscript according to the points raised by the referees.

Concerning referee 1:

We than the referee   for evaluating our manuscript and giving helpful remarks to improve it. Accordingly we have reduced the methodological details and agree with the referee that the manuscript is now more readable.

Kind regards,

Beatrice Bachmeier on behalf of all co-authors

Reviewer 2 Report

I recommend this article “How do German General Practitioners manage Long-/Post-COVID? A qualitative study in primary care” to be published in the journal. Here are some suggestions:

1: “COVID-19 infection (line 136, 186)” should corrected to “SARS-CoV-2 infection”.

2: Please check references carefully.

3: The “Abstract” of the manuscript should be more concise. Please make the necessary correction.

4: In line 45-48, Please add the following reference (DOI: 10.3389/fpubh.2022.908757).

5: The paper is generally well structured. However, there are several review articles with the identical criteria and focus that are recently published. What motivated the authors to prepare the manuscript.

Author Response

Dear Editors,

we have carefully revised the manuscript according to the points raised by the referees.

Concerning referee 2:

We than the referee   for evaluating our manuscript and giving helpful remarks to improve it.

Accordingly we have

(1) substituted "COVID-19 infection" by "SARS-CoV-2 infection".

(2) formatted the referenced to ACS Style (MDPI uses two reference styles, one based on the American Chemical Society (ACS) style and the other following the Chicago style.)

(3) Shortened the abstract to make it more concise

(4) added reference (DOI: 10.3389/fpubh.2022.908757).

(5) About our motivation we have written a paragraph at the end of the introduction section: ("Although guidelines are gradually being developed for the diagnosis and treatment of Long-/Post-COVID, much remains unclear, and existing guidelines for general practitioners and medical specialists on how to diagnose and treat Long-/Post-COVID are far from comprehensive. We conducted a qualitative study to find out how German general practitioners (GPs) deal with this situation, what problems they experience when managing such patients, and how they solve problems associated with the diagnosis and treatment of the condition.")

Kind regards,

Beatrice Bachmeier on behalf of all co-authors

Reviewer 3 Report

The current article by Beatrice E. Bachmeier and al., reviews how do German general practitioners manage Long-/Post-2 COVID. A qualitative study in primary care. The title of the paper is in line with the body of the manuscript. However the data obtained are very very small and unclear. The court is very small and the authors did not provide great and interesting results . I believe that the article can not be accept for this review

Author Response

We have clearly underlined the low number of interviews as significant weakness of the study. Nevertheless our conclusions drawn underline the extensive need for further research to develop and test pharmacological and non-pharmacological interventions, and strategies to prevent the occurrence of Long-/Post-COVID after an acute SARS-CoV-2 infection.

Round 2

Reviewer 3 Report

The corrections made have improved the article even if the data provided are always very very small. The bibliography has also been improved and the article has become more fluid. Despite its limitations I believe that now the article can be accept.